nanotechnology

carbon nanofibres, molybdenum carbides, hybrid materials

**Author for correspondence:**
Mojca Vilfan
e-mail: mojca.vilfan@ijs.si

This article has been edited by the Royal Society of Chemistry, including the commissioning, peer review process and editorial aspects up to the point of acceptance.

# Growth of carbon nanofibres on molybdenum carbide nanowires and their self-decoration with noble-metal nanoparticles

Damjan Vengust[1], Mojca Vilfan[1,2] and Aleš Mrzel[1]

[1]J. Stefan Institute, Jamova 39, 1000 Ljubljana, Slovenia
[2]Faculty of Mathematics and Physics, University of Ljubljana, Jadranska 19, 1000 Ljubljana, Slovenia

MV, 0000-0002-8742-9285

High specific surface area makes carbon nanofibres suitable for catalyst support. Here we report on optimization of carbon nanofibre (CNF) growth on molybdenum carbide nanowires (MoCNW) by direct carburization of $Mo_6S_2I_8$ nanowire bundles. Typical CNFs obtained by this method are several hundreds of nanometres long at a diameter of 10–20 nm. We show that nanofibre growth does not depend on the initial morphology of the nanowires: nanofibres grow on individual bundles of MoCNW, on dense networks of nanowires deposited on silicon substrate, and on free-standing nanowire foils. We find that carbon nanofibres remain firmly attached to the nanowires even if they are modified into $Mo_2C$ and further into $MoS_2$ nanowires. The method thus enables production of a novel hybrid material composed of $MoS_2$ nanowires densely covered with carbon nanofibres. We have additionally shown that the obtained CNFs can easily be self-decorated with platinum nanoparticles with diameters of several nanometres directly from water solution at room temperature without reducing agents. Such efficient synthesis and decoration process yield hybrid platinum/CNF/ molybdenum-based NW materials, which are a promising material for a wide range of possible future applications, including sensitive sensorics and improved catalysis.

## 1. Introduction

The combination of organic and inorganic components in one-dimensional (1D) nanomaterials, such as nanowires, nanotubes and nanorods, represents a very promising concept for developing new functional materials. A combination of organic and inorganic

parts improves optical, electrical and functional properties of the 1D nanomaterials and thus opens a wide range of different applications, including catalysis, nanoelectronics and nanosensorics.

So far, a variety of methods have been proposed and used for fabrication of different 1D organic–inorganic hybrid nanomaterials. These methods include electrospinning [1–3], 1D conjugation of nanoparticles [4,5] and most frequently template-directed synthesis [6]. The simplicity of the template-directed synthesis enables direct transfer of a desired topology in a variety of systems, including channels within solid materials [7,8], structures that were self-assembled from surfactants or block copolymers [9–12], biological superstructures [13–15], or hybridized 1D objects [16–18].

There have been only a few reports on fabrication techniques of hybrid 1D inorganic–organic nanomaterials composed of inorganic nanowires and carbon nanofibres (CNFs). One example of a hybrid material with improved electrochemical performance is single crystalline Ge nanowires (NWs) coated with a thin layer of amorphous carbon. They were grown on CNFs and prepared by an *in situ* vapour–liquid–solid process [19]. Thermal treatment of $Cu_2(NO_3)(OH)_3$ on the surface of CNF led to formation of mesoporous CuO/CNF coaxial shell–core nanowires that have proven to be suitable as anodes for lithium ion batteries [20]. Furthermore, SiNWs with CNF branches were prepared by coating a Ni thin film, followed by the growth of CNF. The obtained hybrid material showed improved stability and improved cycle performance [21].

Besides their interesting physical properties, carbon nanofibres have been long known as catalyst support medium [22–24]. One of the main obstacles for using CNFs in larger catalytic reactors is their small size, which can result in reactor loading problems and pressure drop. To overcome these hindrances, CNFs are often grown on a macroscopic surface with a predefined form, which allows the conservation of their advantages while suppressing diffusional phenomena [25]. Delicate optimization of carbon surface chemistry is required to achieve optimal interaction between the support and metal precursor. For example, fuel cell electrodes were obtained by deposition of platinum nanoparticles on relatively low surface area CNFs by reduction of platinum complexes with different reducing agents [26]. CNF-supported palladium nanoparticles catalyst for Heck reaction was fabricated by combining electrospinning, gas-phase hydrogenation reduction and subsequent calcination [27]. Sari & Ting reported on direct growth of $MoS_2$ on vapour grown CNFs. The obtained hybrid material exhibited improved electrical conductivity [28].

In this work, we present an original synthetic route for growing carbon nanofibres on molybdenum-based nanowires, which can additionally be functionalized with platinum nanoparticles. We used bundles of $Mo_6S_2I_8$ nanowires, synthesized directly from the elements, as starting material. These bundles have already proven to be excellent template precursors for bulk production of several types of molybdenum-based nanowires and nanotubes, including Mo nanowires [29], $MoS_2$ nanotubes and peapods [30,31], $MoO_{3-x}$ nanowires [32] and superconducting MoN nanowires [33]. Recently, we reported on fabrication of different phases of molybdenum carbide nanowires, including a novel hybrid material: molybdenum carbide nanowires covered by carbon nanofibres [34]. In that paper, we reported on formation of nanofibres, which was difficult to control and highly sensitive to specific reaction conditions. Correspondingly, the fabrication of nanofibres on nanowire surface was difficult to reproduce and often only molybdenum carbide nanowires without nanofibres or with a low carbon nanofibre surface density were obtained.

Here we report on an efficient and reproducible process leading to formation of hybrid material composed of densely grown carbon nanofibres on the surface of molybdenum carbide nanowires. CNF lengths were up to several hundreds of nanometres and the nanofibre diameters measured up to a few tens of nanometres. The nanofibres increased the material surface area by a factor of 15: from the initial $10\, m^2\, g^{-1}$ in $Mo_6S_2I_8$ nanowires to over $150\, m^2\, g^{-1}$ in the obtained hybrid material.

# 2. Material and methods

## 2.1. Synthesis

Hybrid materials were obtained by transformation of $Mo_6S_2I_8$ nanowire bundles. First these precursor bundles were synthesized in a single-step reaction as reported previously [31]: they were prepared in an evacuated and sealed quartz ampoule directly from elements. Molybdenum, sulfur and iodine were mixed in a molar ratio of $6:2:8$ and heated to 1040°C for 3 days in a horizontal tube furnace. For most of the carburization experiments, bundles of $Mo_6S_2I_8$ nanowires in the form of textile-like material that formed on the ampoule surface were used as starting material [34]. The same material

was also dispersed in isopropanol (5.0 mg in 100 ml) and deposited by airbrush onto silicon substrate, which resulted in formation of nanowire networks. Evaporation of isopropanol during the spraying was accelerated by heating the substrate to approximately 50°C. The third form of $Mo_6S_2I_8$ was vertically oriented nanowire bundles in free-standing foils grown on the walls of the quartz tube. We synthesized them in a single-step reaction directly from elements like before; however, this time we imposed a temperature gradient between the ends of the sealed quartz tube [35].

Subsequent carburization was performed in an open quartz tube (volume of around 700 cm$^3$ with gas inlet and outlet connections), which was placed horizontally into a single-zone furnace at 710°C. Mass of the starting textile-like $Mo_6S_2I_8$ was 15 mg. We performed carburization in a gas mixture comprising ethane ( purity > 99.99%), argon (purity 5.0) and hydrogen ( purity > 99.99%). Total flowing rate of the gas mixture passing through the quartz tube was around 70 cm$^3$ min$^{-1}$ containing a fixed flow of argon (50 cm$^3$ min$^{-1}$). The flow of hydrogen was in most of the experiments set at 15 cm$^3$ min$^{-1}$ and the flow of ethane at 3 cm$^3$ min$^{-1}$. During the optimization of the carburization parameters, the flow of ethane was changed from 1.0 to 7.5 cm$^3$ min$^{-1}$ and controlled by mass flow controllers (MKS). In the previous paper [34], the temperature heating rate was constant at 7.4 K min$^{-1}$. We now varied the heating rate from 5 to 20 K min$^{-1}$.

Final reduction of the hybrid material from $Mo_2C/MoC$ to $Mo_2C$ phase was conducted in a constant flow of argon/hydrogen mixture (40 and 5 cm$^3$ min$^{-1}$ flows, respectively) for additional 3 h. Sulphurization was done at 800°C for 3 h in flowing argon gas containing 1% of $H_2S$ and 1% of $H_2$ at a flow of 15 cm$^3$ min$^{-1}$.

## 2.2. Decoration with Pt-nanoparticles

Decoration with noble-metal nanoparticles was performed at room temperature without any additional reducing agents. In a typical experiment, 5 mg of nanowire material was dispersed in 100 ml ultrapure water (Millipore) in ultrasonic bath. Water solution of noble-metal chloride complex $Na_2PtCl_4$ and $Na_2PtCl_6$ (5 ml of 2.5 mM) was added dropwise while mixing with a magnetic stirrer. After 1 h of stirring, material was isolated by centrifugation at 2000$g$ for 20 min and subsequently washed several times in ultrapure water.

## 2.3. Characterization

Materials at different stages of synthesis, carburization, sulphurization and noble-metal decoration were observed and analysed by scanning electron microscope (SEM, Jeol JSM-7600F) and high-resolution transmission electron microscope (HR-TEM, Jeol JEM-2100F, 200 keV), equipped with energy dispersive X-ray spectrometer (EDXS, JED-2300T analyser, Jeol). TEM samples were prepared from nanowire dispersions immediately after sonication by deposition onto a copper grid. X-ray powder diffraction (XRD) was performed at room temperature using Bruker AXS D4 Endeavor diffractometer with Cu-K$\alpha_1$ radiation and Sol-X energy-dispersive detector within angular range 2° from 10° to 70° with a step size of 0.02° and a collection time of 2 s at a rotation of 6 r.p.m. Samples were determined by comparison of the XRD spectra with the Joint Committee on Powder Diffraction Standards (JCPDS) files. Brunauer–Emmett–Teller (BET) analysis was performed with Gemini 2370 V5 instrument with the samples being dried overnight at 110°C. With adsorption of $N_2$ gas on a solid surface of tested material (relative pressure in the range 0.02–0.30), the surface area was determined in the units of m$^2$ g$^{-1}$.

# 3. Results and discussion

## 3.1. Precursors

Morphology of $Mo_6S_2I_8$ nanowire bundles, which were used as starting material, was first observed by SEM. Figure 1a shows the textile-like appearance of the material that served as precursor in the majority of the experiments. A large inter-bundle separation is observed with a low degree of agglomeration. Measured bundle lengths are up to several hundreds of micrometres with diameters from several tens up to several hundreds of nanometres. Such a morphology is very suitable for diffusion of gases through the network towards individual nanowires. Furthermore, the morphology enables relatively undisturbed growth of carbon nanofibres on the surface of molybdenum carbide nanowires. Figure 1b shows SEM image of vertical $Mo_6S_2I_8$ nanowires in the form of a free-standing foil. The foil was

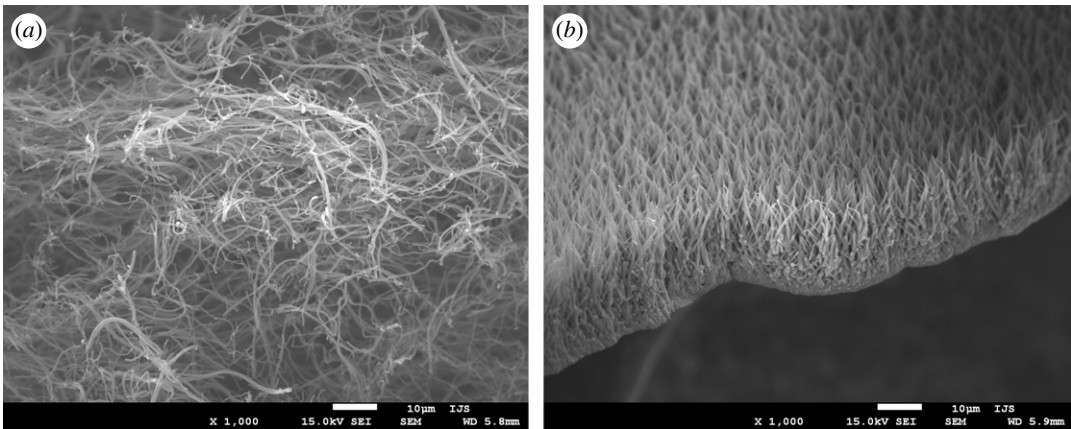

**Figure 1.** SEM images of starting $Mo_6S_2I_8$ nanowires: (*a*) textile-like form of nanowire bundles; (*b*) free-standing foil of nanowire bundles, which was obtained from the lower temperature end of the transport reaction ampoule.

obtained at the lower temperature end of quartz ampoule at transport reaction conditions. The dense back side of the foil shows metallic lustre, while the front side is black due to strong absorption and high porosity. These nanowire bundles are more uniform than in the first case, with a typical diameter of around a hundred nanometres and length of a few tens of micrometres. We performed XRD on both samples and the results, which are consistent with previously published data [36], confirm that the obtained material is indeed $Mo_6S_2I_8$.

## 3.2. Carbon nanofibre growth

In our previous work, we reported that the formation of carbon nanofibres from $Mo_6S_2I_8$ nanowires was very sensitive to reaction conditions and the carburization process often failed to yield carbon nanofibres [34]. The first task was thus to determine the reaction conditions that would result in reproducible bulk synthesis of carbon nanofibres. The starting point of the research was focused around the size of the catalyst particles, as the particle size plays an important role in catalytic decomposition of carbon source. For successful synthesis of CNFs, the particle size must be comparable to the nanofibre diameter, which is of the order of ten nanometres. We speculated that the key parameter that could influence the size of the catalyst particles, which are formed during the carburization of $Mo_6S_2I_8$ nanowire bundles, is the furnace heating rate. We therefore changed the furnace heating rate from 5 to 20 K min$^{-1}$ at a chosen flow of ethane (3 ml min$^{-1}$) and hydrogen (15 ml min$^{-1}$). The reactions were repeated several times using precursor material from different batches in order to exclude possible influence of starting materials. Low-magnification SEM images confirm the conservation of nanowire morphology. High-magnification SEM images, however, reveal a significant difference between obtained materials, depending on the furnace heating rate (figure 2). At a slower heating rate (figure 2*a*), no nanofibres were observed on the nanowire surface. By increasing the heating rate to 10 K min$^{-1}$ and keeping all other parameters unaltered, individual protrusions could be observed on the nanowire surface (figure 2*b*). Their density is fairly low and the majority of the protrusions is short. By further increasing the heating rate (to 15 K min$^{-1}$, figure 2*c*), one observes elongated nanofibres growing on the nanowire surface. At the maximal heating rate of the available furnace (20 K min$^{-1}$, figure 2*d*), the surface density of nanofibres and their length increase further. Length of individual fibres was found to be over 100 nm at a diameter of a few tens of nanometres. BET measurements show that the nanofibres increased the material surface area by a factor of 15: from the initial 10 m$^2$ g$^{-1}$ in $Mo_6S_2I_8$ nanowires to over 150 m$^2$ g$^{-1}$ in the hybrid material.

The samples produced at the highest heating rate, which yields the highest surface density and longest carbon nanofibres (figure 2*d*), were also observed with TEM. Obtained TEM images (figure 3*a,b*) reveal a detailed structure of the material, showing very prominent nanoparticles at the tips of carbon nanofibres. We analysed around 80 such nanoparticles and determined the average diameter of $13 \pm 4$ nm. The largest observed nanoparticle at the tip of a nanofibre had a diameter of around 26 nm. High-resolution TEM images of these nanoparticles were also acquired (figure 3*c*) with a clearly visible crystal structure. Interplanar distance was determined by fast Fourier transform (FFT) and the obtained value of 0.26 nm corresponds to planes 012 and 020 in $Mo_2C$ crystals (figure 3*d*). Observation of each carbon nanofibre

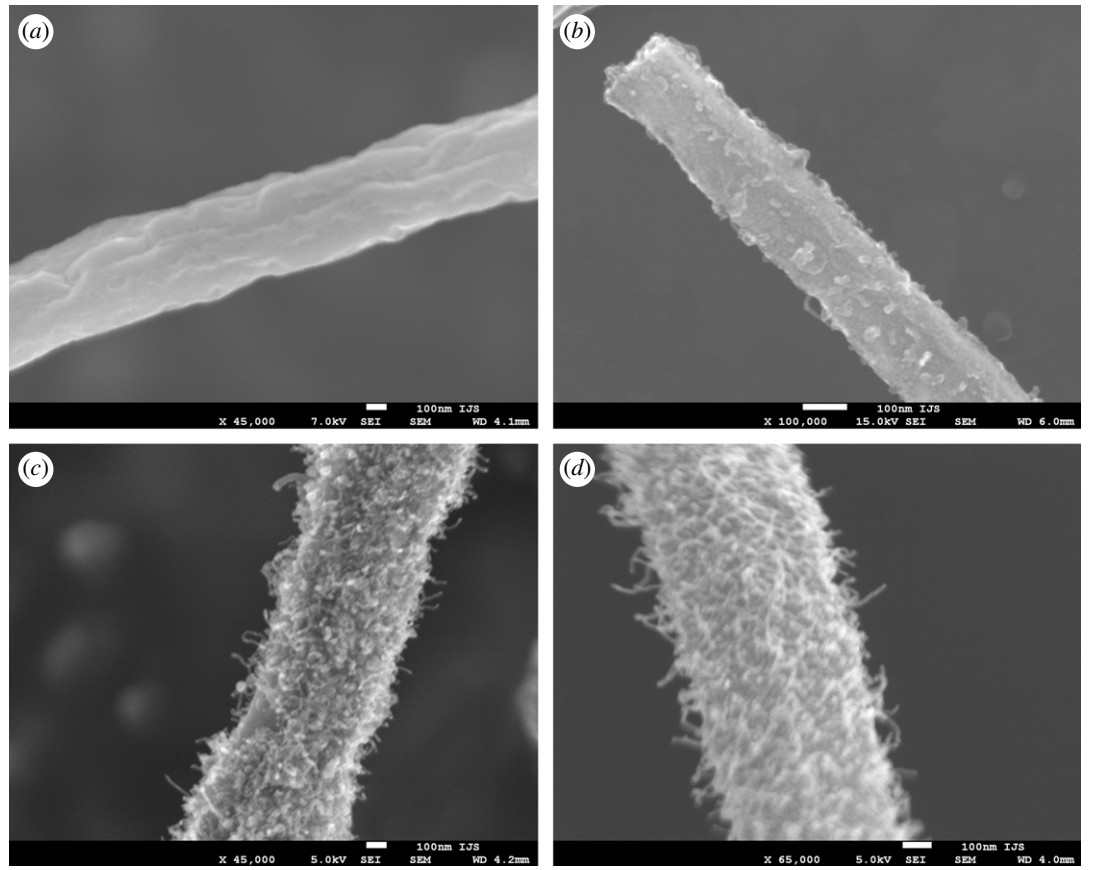

**Figure 2.** SEM images of carbon nanofibres grown on molybdenum carbide nanowires at different heating rates: (*a*) 5 K min$^{-1}$; (*b*) 10 K min$^{-1}$; (*c*) 15 K min$^{-1}$; (*d*) 20 K min$^{-1}$.

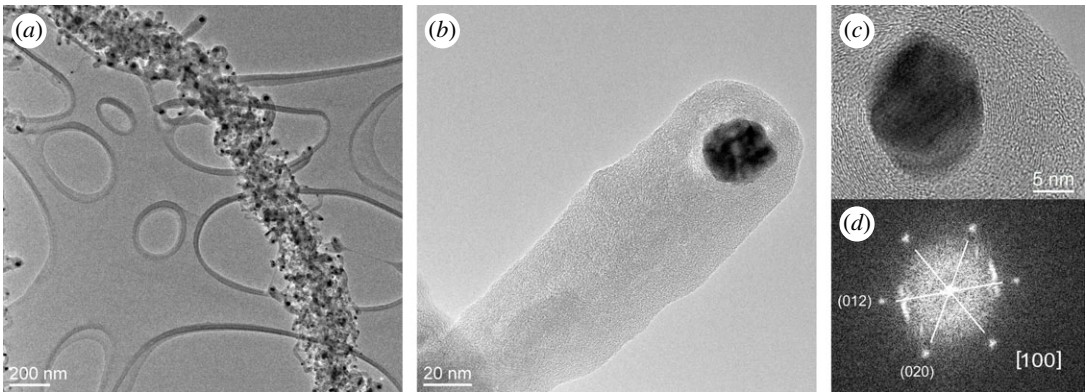

**Figure 3.** TEM images of carbon nanofibres grown on molybdenum carbide nanowires: (*a*) nanowire densely covered with nanofibres; (*b*) individual carbon nanofibre with molybdenum carbide grain at the tip. (*c*) HR-TEM image of the grain, and (*d*) FFT of the image, confirming the composition of the grain as Mo$_2$C.

having a molybdenum carbide grain at the tip is consistent with our previous reports [34] and is crucial in understanding the formation of carbon nanofibres.

Crystal structure of the obtained hybrid material was determined by XRD (figure 4). Spectrum of the initial Mo$_6$S$_2$I$_8$ nanowires is shown as black line and spectrum of the transformed molybdenum carbide as red line. The latter shows characteristic peaks of MoC (JCPDS card no. 08-0384, red triangles) with an additional broad peak observed between 20° and 30°, indicating the presence of carbon in the form of graphite (JCPDS card no. 41-1487, blue star). A comparison with SEM (figure 2) and TEM (figure 3) indicates that the majority of the detected carbon is in the form of nanowires, which is consistent with [37]. Further reduction (green line) of the hybrid material leads to a practically pure Mo$_2$C phase (JCPDS card no. 35-0787, green circles) with a broad peak, which again indicates the presence of layered carbon (JCPDS card no. 41-1487, blue star), predominantly in the form of nanowires.

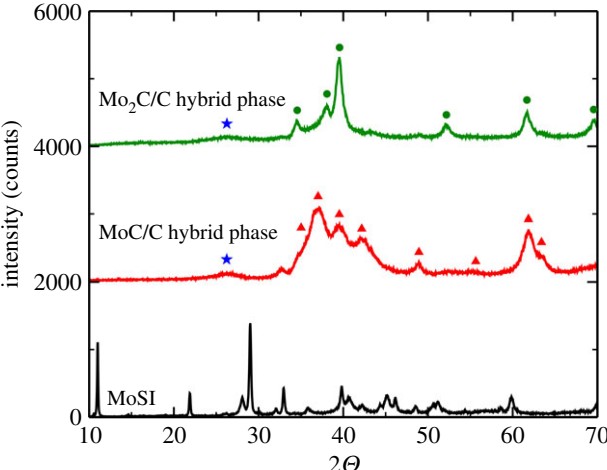

**Figure 4.** XRD spectra of initial $Mo_6S_2I_8$ (black line), molybdenum carbide with carbon nanofibres (red line), and obtained $Mo_2C$ with carbon nanofibres (green line). The peaks annotated with red triangles correspond to JCPDS card no. 08-0384, green circles to JCPDS card no. 35-0787 and blue stars to JCPDS card no. 41-1487.

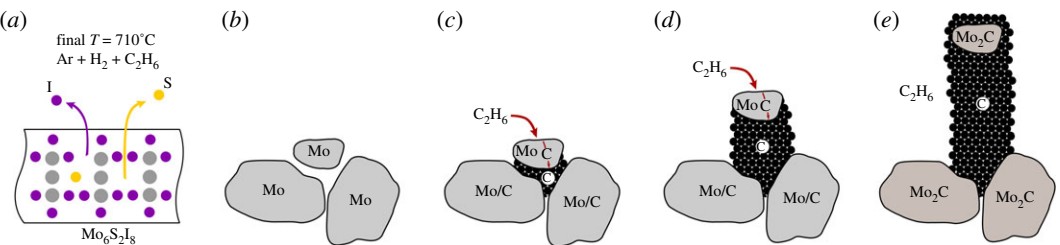

**Figure 5.** Schematic representation of carbon nanofibre growth: (a) $Mo_6S_2I_8$ nanowires are heated causing the release of iodine and sulfur atoms; (b) molybdenum nanowires form, which are composed of individual grains; (c) molybdenum grains act as catalysts and carbon diffuses through the grains; (d) deposition of carbon takes place until Mo grains gradually change to $Mo_2C$ and the growth stops.

## 3.3. Mechanism of carbon nanofibre growth

The appearance of molybdenum carbide grains at the nanofibre tips together with the strong effect of the furnace heating rate lead to the following explanation for the growth mechanism (figure 5): it is known that $Mo_6S_2I_8$ nanowires, when heated to 710°C in mixture of argon and hydrogen without ethane, transform into molybdenum nanowires [29]. Fast release of iodine and sulfur atoms during the nanowire hydrogenation (figure 5a) leads to appearance of pores on the nanowire surface and the crystallization of molybdenum atoms results into formation of grains (figure 5b). We have reported previously that the grains are about 3–50 nm in size [29]. We now observe that the grain size depends strongly on the furnace heating rate: at slow heating rates, the decomposition is slow and rather large molybdenum particles are formed ($22 \pm 7$ nm), which further merge into clusters. On the other hand, faster heating results in formation of smaller particles with an average diameter of $13 \pm 4$ nm. When ethane is included to the reaction, carbon dissolves and diffuses through molybdenum grains until it is deposited on the other side in the form of graphite (figure 5c) [38,39]. Carbon deposition and nanofibre growth continue and the nanoparticles, which serve as catalyst seeds for growing carbon nanofibres, are lifted away from the surface (figure 5d). Molybdenum grains gradually convert into stable $Mo_2C$, through which the carbon can no longer diffuse and the nanofibre eventually stops growing (figure 5e). The $Mo_2C$ grains, which are observed at the tip of each nanofibre, are additionally covered with an approximately 20 nm thick layer of carbon.

If the heating rate is low, initial molybdenum grains are fairly large and they remain bound to the nanowire surface as the strong metal/metal interactions prevent them from being lifted away from the surface. During the carburization, these bound molybdenum particles are further transformed into a mixture of different molybdenum carbide phases. The resulting molybdenum carbide nanowires are composed of grains and grain clusters with a typical size of several tens of nanometres and partially covered with a thin layer of amorphous carbon, but no carbon nanofibres are observed.

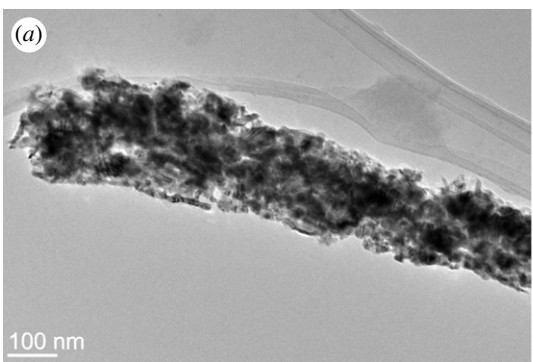
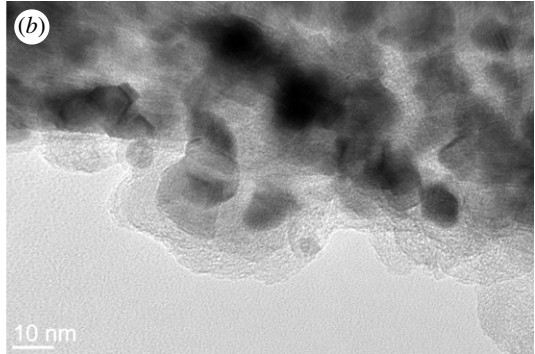

**Figure 6.** TEM images of material obtained by heating $Mo_6S_2I_8$ without hydrogen: (*a*) no nanofibres are present; (*b*) individual molybdenum carbide grains, covered with a layer of amorphous carbon, are observed.

Consistent with the experimental observation, we determine the furnace heating rate of $20\,K\,min^{-1}$ as being the most suitable condition for nanofibre growth due to a fast reaction time and consequently small molybdenum grain sizes. We therefore used this value for synthesis of materials used in further experiments.

## 3.4. Influence of gas composition

Next we heated $Mo_6S_2I_8$ nanowires in the presence of ethane but without hydrogen. We find that the main difference is that no nanofibres were formed on the nanowires. We observe, however, nanowires composed of individual molybdenum carbide grains, which were covered with a thin layer of amorphous carbon (figure 6). These findings are consistent with the proposed mechanism in which molybdenum particles act as catalysts and thus play a crucial role in the formation of nanofibres. The absence of hydrogen allows for a rapid transformation of molybdenum into $Mo_2C$, thus stopping the diffusion, deposition and fibre growth. Originally, the presence of hydrogen, which is a strong reducing agent, slowed down the transformation of the active catalytic particles into molybdenum carbide and gave nanofibres enough time to grow.

The observations led to the conclusion that the efficient nanofibre growth requires not only formation of suitable molybdenum nanoparticles, which is achieved by sufficiently fast decomposition of nanowires, but also proper molar ratio between reacting gases (hydrogen and ethane). We therefore performed a series of experiments with changing molar ratio between ethane and hydrogen, while keeping other parameters unchanged (furnace heating rate of $20\,K\,min^{-1}$ at a final temperature of 710°C). In all the experiments flow of hydrogen was fixed at $15\,ml\,min^{-1}$ and ethane flow was changed from zero to $7.5\,ml\,min^{-1}$. SEM images of materials obtained at different molar ratios between hydrogen and ethane are shown in figure 7. First reaction (figure 7*a*) was done with hydrogen only without ethane and pure molybdenum nanowires are obtained, as reported previously [29]. Already at $1\,ml\,min^{-1}$ ethane flow (molar ratio 15 : 1), nanowire surface was drastically changed. Instead of a porous surface, individual carbon structures with catalytic particles at the tips are seen growing on the surface (figure 7*b*). However, most of the surface is covered with material that does not show typical nanofibre shape. A further increase in ethane flow (3 and $7.5\,ml\,min^{-1}$) significantly increases the density of nanofibres (figure 7*c* and *d*, respectively).

We find that while hydrogen is essential for formation of nanofibres, the level of ethane flow is not the key parameter in nanofibre growth. Nanofibres of proper density and quality efficiently grow in a relatively wide range of gas mixtures composed of hydrogen and ethane. Based on these findings, we decided to use ethane flow of $5\,ml\,min^{-1}$ in further experiments and for bulk production.

## 3.5. Carbon nanofibre on different nanowire morphologies

All of the experiments described so far were performed on the textile-like form of $Mo_6S_2I_8$ nanowires, shown in figure 1*a*. More technologically relevant, however, are nanowires deposited on substrates, which can be used, for example, as sensors or transparent electrodes or solar cells [34]. The nanowires are usually grown on substrates by chemical vapour deposition to improve the contact quality. For some materials, the growth temperature is relatively low and the method can be applied to a wide range of inorganic or organic substrates. The direct transformation of $Mo_6S_2I_8$ nanowires, however, requires relatively high transformation temperatures (up to 710°C) and specific reaction atmosphere, which despite limiting the

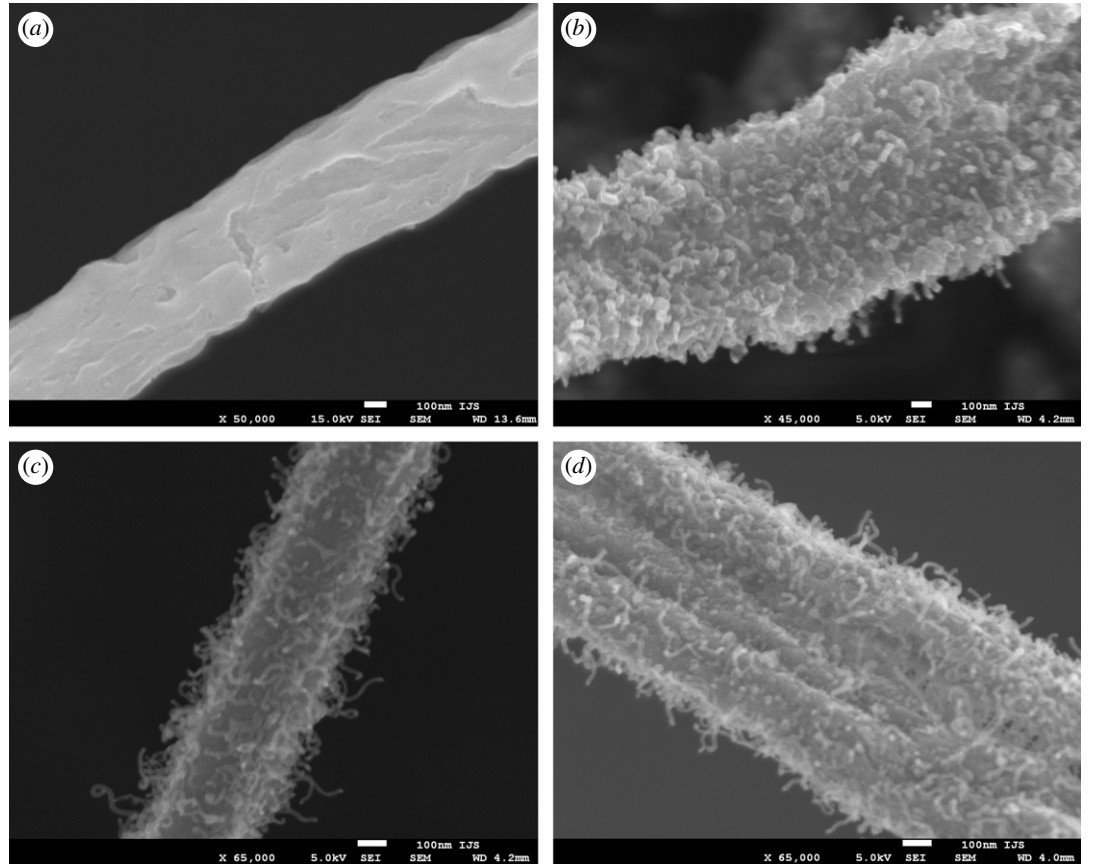

**Figure 7.** Carbon nanofibre growth on nanowires at different molar ratios of gas mixture (hydrogen and ethane): (*a*) molar ratio 15 : 0; (*b*) molar ratio 15 : 1; (*c*) molar ratio 15 : 3; (*d*) molar ratio 15 : 7.5.

choice of substrates, allows the use of technologically important inorganic substrates, such as ceramics, silicon and quartz. We thus sprayed the nanowire solution onto a silicon surface, creating a network of $Mo_6S_2I_8$ nanowires. Following the same direct transformation procedure as with textile-like material (heating rate $20\,K\,min^{-1}$ up to $710°C$, $5\,ml\,min^{-1}$ ethane flow), we obtained networks of molybdenum carbide nanowires densely covered with carbon nanofibres (figure 8*a*–*c*). The morphology of the obtained hybrid material resembles the morphology of deposited starting bundles with lengths up to several hundreds of nanometres. We explain the high density of nanofibres by a low degree of agglomeration, large inter-bundle separation and easy gas diffusion though the loose nanowire network.

The robustness of the adsorption was tested by immersing the deposited and transformed material into water. We find that even after dipping, the transformed nanowires remained tightly bound on silicon surface leading to the conclusion that strong adsorption between the hybrid material and substrate is present.

The third form of $Mo_6S_2I_8$ nanowires are free-standing foils (figure 1*b*). Repeating the same transformation procedure, we obtain nanowires densely covered with carbon nanofibres (figure 8*d*–*f*). This clearly shows the versatility of the method as in all three forms of the initial nanowire bundles, transformation into $Mo_2C/CNF$ hybrid material was successful.

## 3.6. Carbon nanofibre on $MoS_2$ nanowires

Following our previous use of molybdenum-based nanowires as precursors for bulk production of a variety of different nanowires [30–33], the next goal was to test if $Mo_2C/CNF$ could be further transformed into similar molybdenum-based hybrid materials, while preserving the nanowire morphology and carbon nanofibres on the surface. We chose transformation into $MoS_2$ as the material has several important functional properties, including catalytic properties in several reactions and, to the best of our knowledge, there are no reports on carbon nanofibre growth on $MoS_2$ nanowires so far. The only requirement upon the $Mo_2C$ to $MoS_2$ transformation was to preserve the carbon nanofibre structure. After 3 h at $800°C$ in a flowing argon gas containing 1% of $H_2S$ as sulphurization

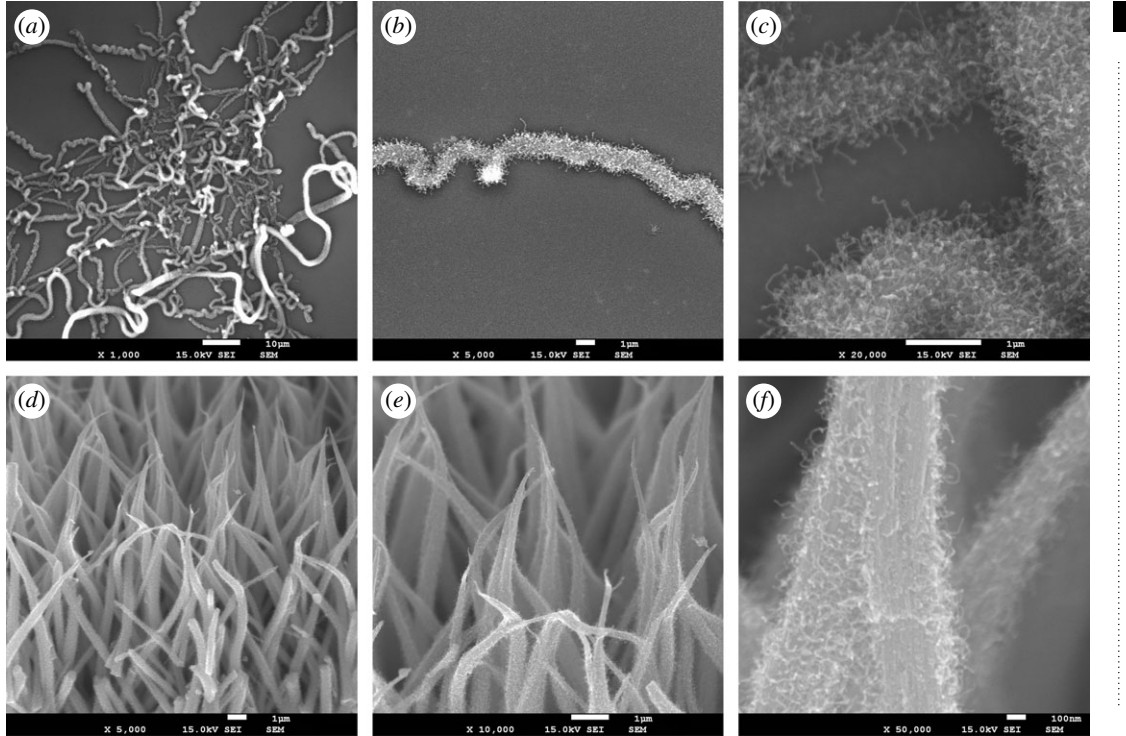

**Figure 8.** SEM images of molybdenum carbide nanowires with visible carbon nanofibres: (*a–c*) nanowires deposited onto a silicon substrate and (*d–f*) free-standing foil of vertically aligned nanowires.

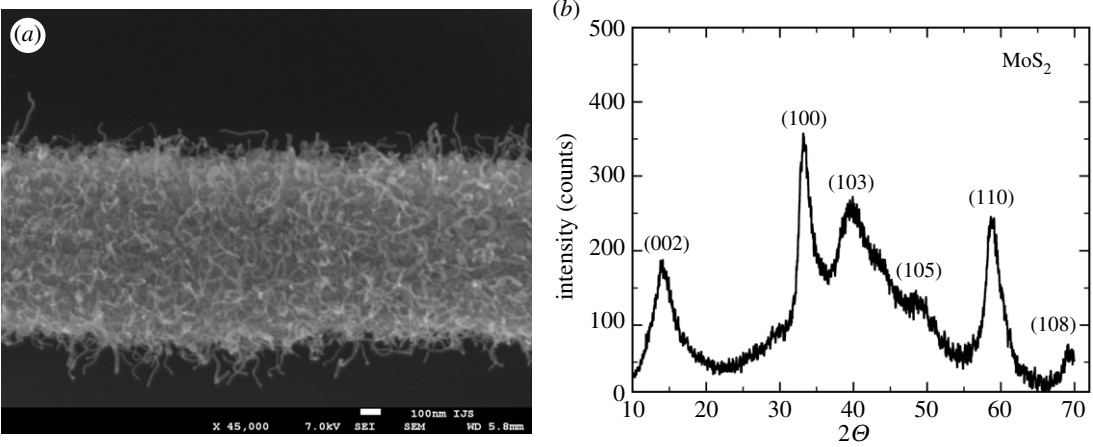

**Figure 9.** MoS$_2$ nanowires with carbon nanofibres: (*a*) SEM image showing the preserved carbon nanofibres and (*b*) XRD confirming the transformation into MoS$_2$. The annotated peaks correspond to JCPDS card no. 77-1716.

agent and 1% of H$_2$ as additional reductant to prevent oxidation of carbon, the structure of the Mo$_2$C/CNF precursor material remained virtually unchanged. SEM image of the obtained material shows successful transformation and preservation of morphology with nanofibres at the surface (figure 9*a*), while XRD spectrum of the MoS$_2$/CNF hybrid material (figure 9*b*) looks notably different from the spectrum of the starting Mo$_2$C/CNF material: the characteristic peaks of Mo$_2$C (figure 4, green line) disappear and the majority of peaks in the observed spectrum can be assigned to MoS$_2$ (JCPDS card no. 77-1716). The remaining broad peak between 20° and 30° indicates the preservation of fibres containing carbon structure (JCPDS card no. 41-1487).

## 3.7. Decoration of carbon nanofibre with Pt-nanoparticles

We further demonstrate functionalization of Mo$_2$C/CNF material. We tested the decoration with platinum nanoparticles by using the method that previously proved highly efficient for decoration of

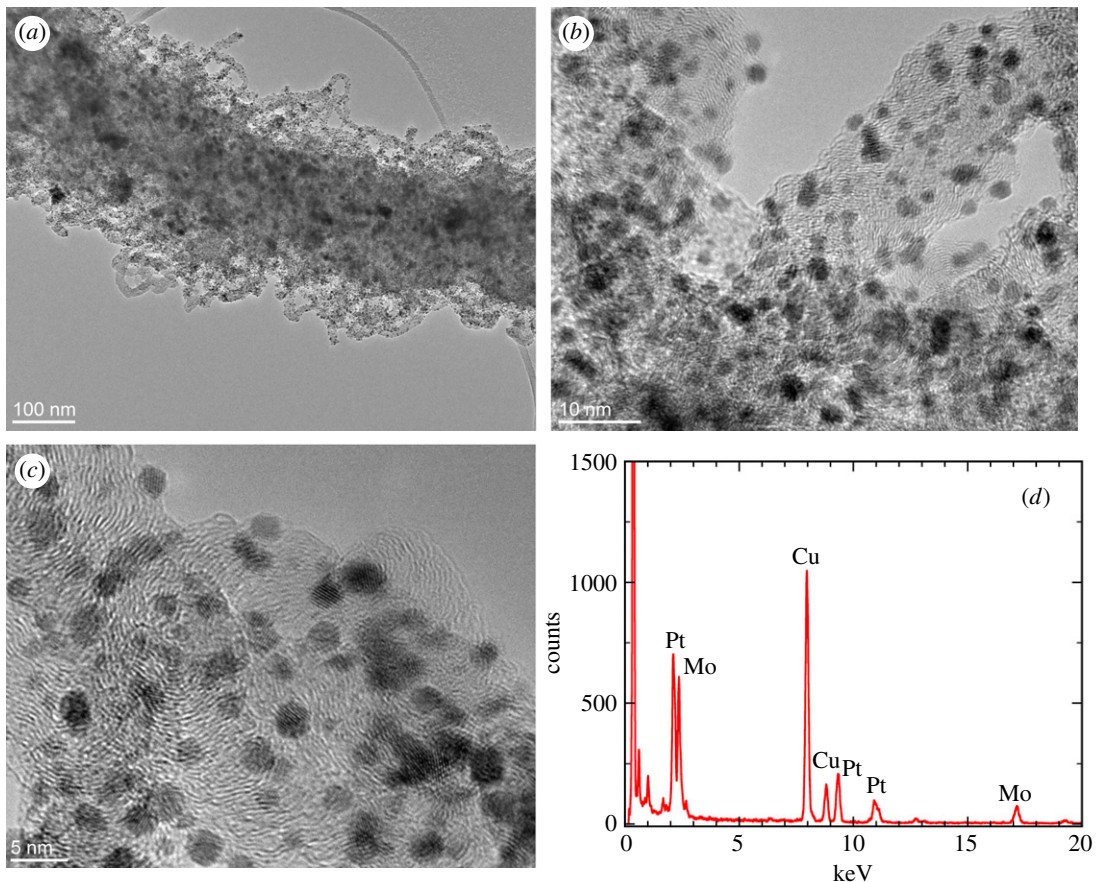

**Figure 10.** Carbon fibres decorated with platinum nanoparticles on $Mo_2C$ nanowires: (a) TEM image showing uniform decoration with nanoparticles; (b) a close-up of carbon nanofibres when $Na_2PtCl_4$ complex was used for decoration; (c) a close-up of carbon nanofibres when $Na_2PtCl_6$ complex was used for decoration; (d) EDS spectrum confirming the presence of platinum.

$MoS_2$ nanotubes [40]. When directly decorating the hybrid materials in aqueous dispersion, we find that the self-decoration of nanofibres with platinum nanoparticles occurs quite rapidly (figure 10a). Already after a few minutes, the carbon nanofibres were decorated with nanoparticles and the supernatant after centrifugation of the samples remained transparent. To achieve a sufficiently homogeneous decoration, the salt solutions were added dropwise into the dispersion of the hybrid material. The main obstacle for efficient and rapid decoration was a relatively low dispersibility of the hybrid material in water. A sufficiently homogeneous dispersion of hybrid materials that would lead to decoration was obtained only after 3 h of vigorous stirring. We believe this is due to a large surface area of the nanowires covered by carbon.

Interestingly, two different sources of platinum ions dissolved in water ($Na_2PtCl_4$ and $Na_2PtCl_6$, figure 10b and c, respectively) yield similar decoration. TEM images reveal an even distribution of platinum nanoparticles over the whole $Mo_2C$ nanowire. The particle size is found to be rather uniform in the range of 2–4 nm. Chemical composition of the obtained nanoparticles was examined by small volume elemental analysis EDXS (figure 10d) and the presence of platinum proven. The platinum nanoparticles notably appear on the surface of carbon nanofibres. This clearly distinguishes them from $Mo_2C$ nanoparticles, which are formed during the reaction and are not only significantly larger but also covered with a thin layer of carbon. The described simple method can thus be used for obtaining highly and uniformly decorated carbon nanofibres with noble-metal nanoparticles. The efficiency of this method was demonstrated on platinum, which makes the decorated hybrid material most attractive candidate for possible future applications as catalyst.

## 4. Conclusions

In this work, we present a reproducible and efficient way for growing carbon nanofibres and production of hybrid molybdenum-based nanowire/carbon nanofibre materials. We have shown the universality of

the method by growing the nanofibres on different forms of nanowires: textile-like, sprayed on a surface, and free-standing foil. The nanofibres remained firmly attached to the nanowire surface even after transformation of the initial nanowires into $Mo_2C$ or $MoS_2$ nanowires. This proves that the applied approach is not limited to carbon nanofibre growth and fabrication of molybdenum carbide-based hybrid materials, but enables fabrication of other molybdenum-based hybrid materials as well.

We have shown that the highly increased surface area of the hybrid material can be decorated and functionalized with noble-metal nanoparticles. Using a straightforward and efficient approach, the decoration of carbon nanofibres was demonstrated on platinum. However, based on previous experience, we believe that the simplicity of the method may allow for decoration with other noble metals as well. The decorated hybrid materials are stable and robust with a high surface area and therefore very promising for possible future industrial applications, sensors, or—as in the case of decoration with platinum—as catalysts.

Data accessibility. Our data are available at Dryad: https://doi.org/10.5061/dryad.931zcrjh2 [41].

Authors' contributions. A.M. designed the study, carried out the synthesis, critically reviewed the manuscript; D.V. performed the microscopy and characterization; M.V. interpreted the results and wrote the manuscript. All authors gave final approval for publication.

Competing interests. The authors declare no competing interests.

Funding. The authors acknowledge the financial support from the Slovenian Research Agency (research core funding P1-0040, P1-0192, P2-0091) and Nanocenter – Center of Excellence in Nanoscience and Nanotechnology.

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
