## [Reviewer comments · Royal Society Open Science]

Review History

RSOS-200783.R0 (Original submission)

Review form: Reviewer 1

Is the manuscript scientifically sound in its present form?

Yes

Are the interpretations and conclusions justified by the results?

Yes

Is the language acceptable?

Yes

Do you have any ethical concerns with this paper?

No

Have you any concerns about statistical analyses in this paper?

No

Recommendation?

Accept with minor revision (please list in comments)

Comments to the Author(s)

Very nice paper and very good results. The nice TEM images would benefit from having an EDS spectra to confirm the nature of the particles. Particularly Fig. 3, it would be nice to confirm the top to be the Mo carbide by EDS. On Fig.10, EDS is provided to confirm the presence of Pt, as it says in the caption. However, which zone of the image was analysed? Was it an average grains or a specific part?

Authors say that the materials are suitable for industrial applications, sensors or catalysts, but no attempt to use them was performed. I think it should, unless authors are able to justify the novelty of these materials in a way that publication is viable even without any tests for their potential applications being performed.

Review form: Reviewer 2

Is the manuscript scientifically sound in its present form?

Yes

Are the interpretations and conclusions justified by the results?

No

Is the language acceptable?

Yes

Do you have any ethical concerns with this paper?

No

Have you any concerns about statistical analyses in this paper?

No

Recommendation?

Major revision is needed (please make suggestions in comments)

Comments to the Author(s)

Authors reported an approach of carbon nanofibre (CNF) growth on molybdenum carbide nanowires (MoCNW) by direct carburization of Mo₆S₂I₈ nanowire bundles and studied many effect factors. It is a valuable work. However, some issues are still to be addressed before being considerate by Journal.

1. BET results are not found in manuscript and it is described in characterization part .
2. Which sample' s TEM images are presented in Figure 3? How to determine the particle was molybdenum carbide in line 52, page 6?
- 3 For Figure 4, authors stated the broad peak observed between 20° and 30° was pyrolytic carbon for red line and another broad peak corresponded to carbon nanofibres for green line. Why? How to confirm that is the different carbon for the same position? Please provide evidence.
4. In Mechanism of CNF growth part, particle size such as 3-50nm and 10-15nm. etc are lack of direct evidences, for example size distribution.
5. How to conform molybdenum nanowires in Fig.7a? be lack of evidence.
6. It is easily found that molybdenum carbide nanowires were composed of Mo₂C nanoparticles. How to distinguish those particles were Mo₂C or Pt? Is there element analysis?
7. Figures should be normalized, e.g. axis labels. Some dimensionless in Y-axis, ect.

Decision letter (RSOS-200783.R0)

Dear Dr Vilfan:

Title: Growth of carbon nanofibres on molybdenum carbide nanowires and their self-decoration with noble-metal nanoparticles
Manuscript ID: RSOS-200783

The editor assigned to your manuscript has now received comments from reviewers. We would like you to revise your paper in accordance with the referee and Subject Editor suggestions which can be found below (not including confidential reports to the Editor). Please note this decision does not guarantee eventual acceptance.

Please submit your revised paper before 27-Aug-2020. Please note that the revision deadline will expire at 00.00am on this date. If we do not hear from you within this time then it will be assumed that the paper has been withdrawn. In exceptional circumstances, extensions may be possible if agreed with the Editorial Office in advance. We do not allow multiple rounds of revision so we urge you to make every effort to fully address all of the comments at this stage. If deemed necessary by the Editors, your manuscript will be sent back to one or more of the original reviewers for assessment. If the original reviewers are not available we may invite new reviewers.

On behalf of the Subject Editor Professor Anthony Stace and the Associate Editor Dr Dattatray Late.

RSC Associate Editor:
Comments to the Author:
(There are no comments.)

RSC Subject Editor:
Comments to the Author:
(There are no comments.)

Reviewers' Comments to Author:
Reviewer: 1

Comments to the Author(s)

Very nice paper and very good results. The nice TEM images would benefit from having an EDS spectra to confirm the nature of the particles. Particularly Fig. 3, it would be nice to confirm the top to be the Mo carbide by EDS. On Fig.10, EDS is provided to confirm the presence of Pt, as it says in the caption. However, which zone of the image was analysed? Was it an average grains or a specific part?

Authors say that the materials are suitable for industrial applications, sensors or catalysts, but no attempt to use them was performed. I think it should, unless authors are able to justify the novelty of these materials in a way that publication is viable even without any tests for their potential applications being performed.

Reviewer: 2

Comments to the Author(s)

Authors reported an approach of carbon nanofibre (CNF) growth on molybdenum carbide nanowires (MoCNW) by direct carburization of Mo₆S₂I₈ nanowire bundles and studied many effect factors. It is a valuable work. However, some issues are still to be addressed before being considerate by Journal.

1. BET results are not found in manuscript and it is described in characterization part .
2. Which sample' s TEM images are presented in Figure 3? How to determine the particle was molybdenum carbide in line 52, page 6?
- 3 For Figure 4, authors stated the broad peak observed between 20° and 30° was pyrolytic carbon for red line and another broad peak corresponded to carbon nanofibres for green line. Why? How to confirm that is the different carbon for the same position? Please provide evidence.
4. In Mechanism of CNF growth part, particle size such as 3-50nm and 10-15nm. etc are lack of direct evidences, for example size distribution.
5. How to conform molybdenum nanowires in Fig.7a? be lack of evidence.
6. It is easily found that molybdenum carbide nanowires were composed of Mo₂C nanoparticles. How to distinguish those particles were Mo₂C or Pt? Is there element analysis?
7. Figures should be normalized, e.g. axis labels. Some dimensionless in Y-axis, ect.

Author's Response to Decision Letter for (RSOS-200783.R0)

See Appendix A.

RSOS-200783.R1 (Revision)

Review form: Reviewer 1

Is the manuscript scientifically sound in its present form?

Yes

Are the interpretations and conclusions justified by the results?

Yes

Is the language acceptable?

Yes

Do you have any ethical concerns with this paper?

No

Have you any concerns about statistical analyses in this paper?

No

Recommendation?

Accept as is

Comments to the Author(s)

In my opinion, the paper is nice and well written and can be accepted as it is. Authors replied well to the comments raised by the referees and the paper largely improved.

Decision letter (RSOS-200783.R1)

Dear Dr Vilfan:

Title: Growth of carbon nanofibres on molybdenum carbide nanowires and their self-decoration with noble-metal nanoparticles

Manuscript ID: RSOS-200783.R1

It is a pleasure to accept your manuscript in its current form for publication in Royal Society Open Science. The chemistry content of Royal Society Open Science is published in collaboration with the Royal Society of Chemistry.

The comments of the reviewer who reviewed your manuscript are included at the end of this email.

Yours sincerely,
Dr Ellis Wilde
Publishing Editor, Journals

On behalf of the Subject Editor Professor Anthony Stace and the Associate Editor Dr Dattatray Late.

RSC Subject Editor
Comments to the Author:
Accept as is

RSC Associate Editor
Comments to the Author:
(There are no comments.)

Reviewer(s)' Comments to Author:
Reviewer: 1

Comments to the Author(s)

In my opinion, the paper is nice and well written and can be accepted as it is. Authors replied well to the comments raised by the referees and the paper largely improved.

Appendix A

Dear Dr Smith,

Thank you for the reply and the accompanying Referees' reports on our manuscript "Growth of carbon nanofibres on molybdenum carbide nanowires and their self-decoration with noble-metal nanoparticles" (RSOS-200783). We have modified the manuscript accordingly to their comments and suggestions. We are now submitting the revised version with a detailed Response to Referees. We are confident that the improved manuscript is suitable for publication in RSOS.

Yours sincerely,

Mojca Vilfan, on behalf of all authors

Response to Referees

We would like to thank both Referees for their positive response and useful comments. We have considered and addressed all their questions and modified the manuscript accordingly. A detailed response with specific modifications is given below.

Reviewer 1:

We thank and appreciate the positive response from the Reviewer.

1. The nice TEM images would benefit from having an EDS spectra to confirm the nature of the particles. Particularly Fig. 3, it would be nice to confirm the top to be the Mo carbide by EDS.

We have acquired EDS spectra from the particles at the tips of the nanofibres and presence of both molybdenum and carbon was confirmed. However, EDS is not enough to determine the exact structure of the observed material. We have therefore taken high-resolution TEM images of the particles at the ends and performed FFT on the obtained particle images. The observed pattern and interplanar distance correspond to Mo₂C structure of the particles. We have added the detailed HR-TEM image with the corresponding FFT image to Figure 3 in the manuscript and adjusted the text accordingly.

2. On Fig.10, EDS is provided to confirm the presence of Pt, as it says in the caption. However, which zone of the image was analysed? Was it an average grains or a specific part?

Energy dispersive X-ray spectrometer (EDXS) is a component of high-resolution TEM. The nominal spot size, in which the spectrum is measured, is 2.4 nm (according to the manufacturer Jeol Ltd.), although ~5 nm is probably a more realistic value for our set-up. Combined with TEM visualisation, we are able to select a single nanoparticle and analyse it for its elemental composition. We have replaced Fig. 10d and are now showing a measurement which was done on a specific part of the nanofibre (away from the nanowire and also away from the tip). Platinum (and carbon) peaks are very prominent, while some background, which includes molybdenum and copper grid, is also seen on the spectrum. We have added the description of EDXS to the Characterisation (section 2c) and an explanation about the point/small volume elemental analysis to the results (section 3g).

3. Authors say that the materials are suitable for industrial applications, sensors or catalysts, but no attempt to use them was performed. I think it should, unless authors are able to justify the novelty of these materials in a way that publication is viable even without any tests for their potential applications being performed.

We have modified the manuscript accordingly and changed the applications to candidates for possible future applications. Our initial statement was based on general suitability of nanomaterials that are densely covered with platinum nanoparticles. We agree with the Reviewer that since no direct proofs for industrial use of the obtained material are provided, we should have avoided such claims.

Reviewer 2:

We thank the Reviewer for the comments, questions and suggestions, which have all been carefully addressed.

1. *BET results are not found in manuscript and it is described in characterization part.*

We present BET results in the Results section, at the end of part (b). The results clearly demonstrate a significant increase in the material surface area of the hybrid material, by a factor of 15. We have also added some measurement parameters to the Characterisation (section 2c).

2. *Which sample's TEM images are presented in Figure 3? How to determine the particle was molybdenum carbide in line 52, page 6?*

The nanowires, presented in Figure 3 are the ones obtained at the maximal heating rate, yielding the highest nanofibre surface density and longest nanofibres, corresponding to Figure 2d. We have added the clarification to section 3b.

As for characterisation of the particles at the ends of the carbon nanofibres, EDS on these particles was not sufficient as only the presence of elements (molybdenum and carbon) was determined. We have therefore added high-resolution TEM images of these particles, in which crystal structure is clearly visible. Interplanar distance was determined by Fast Fourier Transform (FFT) and the obtained separation of 0.26 nm is found to be consistent with observed 012 and 020 planes of Mo₂C crystal. The HR-TEM and FFT images were added to Fig. 3 and the description to section 3b.

3. *For Figure 4, authors stated the broad peak observed between 20° and 30° was pyrolytic carbon for red line and another broad peak corresponded to carbon nanofibres for green line. Why? How to confirm that is the different carbon for the same position? Please provide evidence.*

We agree that the statements are somewhat unclear. The two different spectra both show the peak between 20° and 30°, which corresponds to the same material – carbon in the form of layered graphite. This is also noted in the Figure as blue star and confirmed by the assigned JCPDS card 41-1487, which is the same in both graphs. The XRD, however, only confirms the presence of carbon and a comparison with SEM and TEM images is required to confirm that nanofibres are made of carbon – with atomic layers clearly visible on TEM images (Figures 3 and 10). We have corrected the manuscript accordingly.

4. *In Mechanism of CNF growth part, particle size such as 3-50nm and 10-15nm. etc are lack of direct evidences, for example size distribution.*

The initial values we gave were just a rough estimation of particle sizes. We have now analysed several TEM images in which Mo₂C nanoparticles are visible and the results are the following: The nanoparticles, which are found at the ends of carbon nanofibres have an average diameter of 13 nm +/- 4 nm, with almost 80 particles analysed. The smallest measured particle was 6 nm in diameter and, more importantly, the largest observed had a diameter of 25 nm. No larger particles

were observed. This is an important observation as the particle size determines whether a fibre will grow, and also influences (limits) the carbon nanofibre diameters to a couple of tens of nanometres.

Measuring the size of the particles forming the nanowire is not as straightforward, as individual particles are grown together forming larger clusters. We analysed around 40 particles and found the average size of individual grains to be 22 nm +/-7 nm, however, they are usually found in the form of larger clusters, so talking about particle size is rather difficult. We have added these findings to the manuscript.

5. How to confirm molybdenum nanowires in Fig.7a? be lack of evidence.

Indeed, there was no evidence given. We have added a reference to one of our previous papers [29], in which we describe the transformation of MoSI nanowires into pure Mo nanowires in detail, including thorough characterisation by XRD. Since in this case the same procedure was used (transformation with hydrogen only, without ethane), we are confident that the nanowire shown in Fig. 7a is pure molybdenum nanowire.

6. It is easily found that molybdenum carbide nanowires were composed of Mo₂C nanoparticles. How to distinguish those particles were Mo₂C or Pt? Is there element analysis?

Yes, we have performed point/small volume elemental analysis EDXS on the sample, which allows us to focus on an area of a few nanometres. The graph in Fig. 10d confirms presence of platinum (plus Cu and Mo from the background). Besides, there are several indicators that distinguish between Mo₂C and Pt particles: TEM images clearly show platinum particles on the surface of carbon nanofibre, whereas Mo₂C particles are covered with a thin layer of pure carbon. The sizes of platinum particles are much smaller, only a few nanometres in diameter, whereas Mo₂C are typically up to 5 times larger – they can actually be seen in Fig. 10a in the nanowire whereas the platinum binds to carbon nanofibre surface. We have replaced previous Fig. 10d with one focusing on carbon nanofibre with platinum. Platinum peaks in the spectrum are much more pronounced and much less of molybdenum is observed in the background as in the graph shown previously.

7. Figures should be normalized, e.g. axis labels. Some dimensionless in Y-axis, ect.

We omitted the notation “arbitrary units” and changed it to original “Counts”, as is the standard for presenting XRD data. The absolute values of the peaks are rather unimportant and while we could normalise all the data to a given value, it would be an arbitrarily chosen one and the normalisation would not yield any new information. The data is used only to prove presence of certain materials. We are now presenting both XRD in EDS in the conventional units (Counts as obtained directly from the experiment).